



# 1 Role of atmospheric horizontal resolution in simulating

# 2 tropical and subtropical South American precipitation in

# 3 HadGEM3-GC31

Paul-Arthur Monerie[1], Amulya Chevuturi[1], Peter Cook[1], Nick Klingaman[1], Christopher E. Holloway[2]
[1] Department of Meteorology, National Centre for Atmospheric Science (NCAS), University of
Reading, Reading, UK
[2]Department of Meteorology, University of Reading, Reading, UK
*Correspondence to*: Paul-Arthur Monerie (pmonerie@gmail.com)

**11    Abstract**

We assess the effect of increasing horizontal resolution on simulated precipitation over South America in
a climate model. We use atmosphere-only simulations, performed with HadGEM3-GC31 at three horizontal
resolutions: N96 (~130 km, 1.88° x 1.25°), N216 (~60 km, 0.83° × 0.56°), and N512 (~25 km, 0.35° x
0.23°). We show that all simulations have systematic biases in annual mean and seasonal mean precipitation
over South America (e.g. too wet over the Amazon and too dry in northeast). Increasing horizontal
resolution improves simulated precipitation over the Andes and north-east Brazil. Over the Andes,
improvements from horizontal resolution continue to ~25km, while over north-east Brazil, there are no
improvements beyond ~60km resolution. These changes are primarily related to changes in atmospheric
dynamics and moisture flux convergence. Over the Amazon basin, precipitation variability increases at
higher resolution. We show that some spatial and temporal features of daily South American precipitation
are improved at high resolution, including the intensity spectra of rainfall. Spatial scales of daily



precipitation features are also better simulated, suggesting that higher resolution may improve the
representation of South American mesoscale convective systems.

## 1. Introduction

South America is a large area encompassing tropical, sub-tropical and extratropical climates. The Andes
covers western South America, from South to North, while the eastern part of South America is flatter than
the west. The Amazon basin has high mean rainfall and is covered by a rainforest, while northeastern Brazil
is semi-arid. Several climatic areas are thus often defined to account for the climatic heterogeneity of South
America, with focus specifically on the Andes, the Amazon Basin, north-east Brazil and south-east Brazil
(de Souza Custodio et al. 2017).
Climate models have biases in simulating South American precipitation, partly due to biases in simulating
teleconnections between both Atlantic and Pacific sea-surface temperatures (SSTs), and precipitation over
land (Bombardi and Carvalho 2008; Jung et al. 2011; Yin et al. 2013; Sierra et al. 2015; Coelho et al. 2016;
Koutroulis et al. 2016). At sub-seasonal scales, precipitation variability is associated with the Madden—
Julian Oscillation (MJO) (Grimm 2019). The MJO modulates precipitation over South America, leading to
either anomalously dry or wet conditions over South America, depending on its phase. The MJO also favors
extreme events, leading to droughts and floods (Grimm 2019). At inter-annual scales, the El Niño Southern
Oscillation (ENSO) strongly impacts Amazon precipitation, with El Niño events related to droughts
(Grimm and Silva Dias 1995; Zeng et al. 2008; Marengo et al. 2008, 2011, 2013; Grimm and Tedeschi
2009; Lewis et al. 2011). Variability in the tropical Atlantic Ocean modulates trade easterlies and impacts
precipitation over north-east Brazil (Liu and Juárez 2001; Zeng et al. 2008) and south-east Brazil (Coelho
et al. 2016). On decadal to multi-decadal scales, variability in north-east Brazilian precipitation is tied to
the Atlantic Multidecadal Variability, which is associated with the location of the Atlantic Intertropical
Convergence Zone (ITCZ) (Knight et al. 2006). Brazilian precipitation is also associated with Interdecadal





Pacific Variability (IPV; Power et al. 1999), positive IPV phases reduce precipitation over South America
(Villamayor et al. 2018). Errors in simulating teleconnections from local and remote SST variability leads
to biases in the intensity, position of the ITCZ and the South Atlantic Convergence Zone (SACZ), which
degrade simulated South American precipitation and temperature (Bombardi and Carvalho 2008; Custódio
et al. 2012; de Souza Custodio et al. 2017).
Besides teleconnections, climate variability results from complex local interactions between energy,
precipitation and soil moisture. These feedbacks are particularly strong over interior South America, one
of the "hot spots" in soil moisture—precipitation coupling (Koster et al. 2004; Wei and Dirmeyer 2012).
Variability in recycling accounts for a large fraction of precipitation variability over north-eastern Brazil
and the La Plata Basin (Sörensson and Menéndez 2011). Soil moisture memory influences atmospheric
variability and could affect the development of the South American Monsoon System. Therefore, biases in
simulated South American climate may be partly attributed to biases in local land-atmosphere coupling.
Improving simulated precipitation in climate models may also improve subseasonal-to-decadal predictions,
because the performance of initialised forecasts and free-running models relies on the representation of key
physical processes, such as convection and land-atmosphere feedbacks. For instance, models with the
largest systematic errors produce the lowest precipitation prediction performance (DelSole and Shukla
2010). Jia et al. (2014) showed that the high-resolution version of the GFDL model produces lower biases
and higher skill for seasonal variations of 2-m air temperature and precipitation over South America, than
its lower-resolution counterpart. Therefore, Doblas-Reyes et al. (2013) proposed that increasing spatial
resolution is one of the main challenges for improving predictions.
Horizontal resolutions of Coupled Model Intercomparison Project (CMIP; Taylor et al. 2012; Eyring et al.
2016) models are typically ~150 km, or coarser, in the atmosphere, and ~100 km in the ocean. Important
climate processes, such as atmospheric convection, and mesoscale boundary currents and eddies, have to
be parameterized rather than resolved, which may compromise dynamical processes and dynamics-physics





interactions (Collins et al. 2018). A growing body of evidence shows then that increasing horizontal
resolution can improve some aspects of the simulated climate (Roberts et al. 2018, 2019; among others).
Higher-resolution ocean-atmosphere coupled models outperform lower-resolution models at simulating
SST over coastal upwelling regions, due to a better simulation of near-surface wind and its effect on the
ocean (Shaffrey et al. 2009; Gent et al. 2010; McClean et al. 2011; Delworth et al. 2011; Sakamoto et al.
2012; Small et al. 2014).  Resolution reduces the double ITCZ bias (Delworth et al. 2011) and improves
variability in the El-Niño Southern Oscillation (Shaffrey et al. 2009; Sakamoto et al. 2012; Small et al.
2014) and north Atlantic SSTs (Gent et al. 2010). Jung et al. (2011) and Jia et al. (2014) highlighted that
increased resolution improved simulated South American precipitation and tropical mean precipitation, and
atmospheric circulation. Improved land precipitation is partly due to a better representation of orography
(Gent et al. 2010; Delworth et al. 2011; Sakamoto et al. 2012). Over South America, increasing horizontal
resolution improves the representation of climate patterns (de Souza Custodio et al. 2017), particularly over
the Ocean, over the Atlantic ITCZ and SACZ. Although strongly model and season dependent, high
resolution regional climate models also improve simulated precipitation and temperature over South
America (Falco et al. 2019; Solman and Blázquez 2019). Increased resolution also affects local features,
such as the propagation of mesoscale systems (Vellinga et al. 2016) and local land-atmosphere feedbacks
(Mueller et al. under review).
However, horizontal resolution does not always improve simulated climate. Bacmeister et al. (2013) found
that the high-resolution Community Atmosphere Model (CAM) did not improve simulated South American
rainfall, compared to a lower-resolution configuration. Some simulations exhibit too much warming and
cooling, especially over polar regions where sea ice is not accurately represented (McClean et al. 2011;
Kirtman et al. 2012). Impacts of increased horizontal resolution strongly depend on the range of resolutions
considered, on the region, phenomena and spatial and temporal scales of interest (Jung et al. 2011; Roberts
et al. 2018). Therefore, there is a need to better understand how increasing the horizontal resolution could
benefit simulated South American precipitation.





Accurate predictions and projections of extreme rainfall require realistic simulated precipitation
distributions. However, models exhibit biases in the frequency and persistence of light (<10 mm.day$^{-1}$) and
heavy precipitation (>20 mm.day$^{-1}$) (Sun et al. 2006; Dai 2006; Koutroulis et al. 2016). Errors in
precipitation frequency and intensity are related to biases in the global hydrological cycle, including
evaporation recycling over land (Trenberth 2011; Demory et al. 2014). Improved representations of intense
small-scale events improves precipitation variability in models over parts of South America (De Sales and
Xue 2011). These biases may be partly due to the coarse resolution of CMIP climate models; increased
resolution could improve simulated extreme convective rainfall by enhancing smaller-scale precipitation
features, as shown by Solman and Blázquez (2019) over South America.
High resolution models are costly; if higher resolution produces little or no improvements in model biases,
then computational resources could be used elsewhere, such as in increased ensemble size or adding
initialization dates in forecasting systems, or improved or additional model physics. The European Union's
Horizon 2020 PRIMAVERA project ([www.primavera-h2020.eu](http://www.primavera-h2020.eu)) uses the CMIP6 High Resolution Model
Intercomparison Project (HighResMIP; Haarsma et al. 2016) protocol and aims to develop a new generation
of advanced high-resolution global climate models.
We use PRIMAVERA simulations to evaluate whether increased horizontal resolution improves simulated
South American precipitation. We address three main questions:
- What are the model biases in simulated precipitation over South America?
- Is South American mean precipitation and variability better simulated at higher than at lower resolution?
What is the minimum resolution required to improve the lower resolution biases?
- Are the spatial and temporal organizations of precipitation, better simulated at higher resolution?





The paper is structured as follows: the model, data and methodology are described in Sect. 2. Sect. 3 focuses
on the model's ability to simulate annual and seasonal precipitation mean. We discuss seasonal to
interannual variability in Sect. 4 and daily to sub-seasonal variability and spatial and temporal scales of
precipitation in Sect. 5. A conclusion is given in Sect. 6.





## 2. Data and Methods

### 2.1 HadGEM3-GC3.1

HadGEM3-GC3.1 (hereafter HadGEM3) (Williams et al. 2018) has been run in an atmosphere-only configuration for 1950-2014, forced by HadISST2 daily 0.25° SSTs and sea ice (Rayner et al. 2006). The atmospheric model is the Global Atmosphere 7.1 scientific configuration (Walters et al. 2019), with 85 vertical levels. A common historical forcing is imposed in all simulations, including SSTs, greenhouse gases and aerosols. Three sets of simulations are performed, which only differ by their horizontal resolution and by a stochastic perturbation of their initial conditions: N96 horizontal resolution (~130 km, 1.88° x 1.25°; HadGEM3-GC3.1-LM), N216 horizontal resolution (~60 km, 0.83° × 0.56°; HadGEM3-GC3.1-MM) and N512 horizontal resolution (~25 km, 0.35° x 0.23°; HadGEM3-GC3.1-HM). Three members were performed at each resolution, for a total of 9 simulations.

### 2.2 Observations and reanalysis

To verify the spatial and temporal scales of rainfall, three-hour and daily mean precipitation from HadGEM3 is compared against a high-resolution (0.25° x 0.25°) satellite-derived product for 1998-2017: NOAA CPC Morphing Technique (CMORPH version 1; Joyce et al. 2004). To evaluate time-mean rainfall and sub-seasonal to seasonal variability, we compare HadGEM3 to longer-period, but lower-resolution, gauge-based datasets from the University of Delaware (Willmott et al. 2001) and from the Global Precipitation Climatology Centre (GPCC; Schneider et al. 2014). We assess mean circulation against the NCEP-NCAR reanalysis (Kanamitsu et al. 2002) and ERA-interim reanalysis (Dee et al. 2011).

To assess biases and impacts of the horizontal resolution on mean annual and seasonal precipitation we used monthly data, over 1950-2014, using GPCC and ERA-interim. For daily variance we used GPCC,



over 1982-2014. For the analysis of the spatial scales in precipitation, we used CMORPH, over 1998-2014.
Note that results in mean and variance in precipitation were also assessed with CMORPH, in addition to
GPCC, for a consistency with the spatial scales analysis.

## 2.3 Data interpolation

Differences between HadGEM3 and observations and between HadGEM3 at different horizontal
resolutions are assessed by first interpolating all data to a common 0.5° x 0.5° resolution. Results were
repeated, with data interpolated onto a common coarser resolution, 2.5° x 2.5° grid, showing similar results.
For the analysis of the spatial scales in precipitation, both simulations and observations are interpolated
onto a common lower resolution, N96.

## 2.4 Analysis of Scales of Precipitation (ASoP)

The Analysis of Scales of Precipitation (ASoP; Klingaman et al. 2017; Martin et al. 2017) diagnostics
provide information on the intensity spectra of precipitation, the contribution to total precipitation from
precipitation events of various intensities, the temporal persistence of precipitation and the typical spatial
and temporal scales of precipitation.
The intensity spectra measures intensity distributions by computing the contributions of discrete intensity
bins to the total precipitation for each grid point, to be visualised as maps (at grid scale) or aggregated over
regions into histograms. Spatial scales of precipitation features are measured by dividing the analysis
domain into non-overlapping subregions and computing correlations of each point in the sub-region against
the central grid point, then averaging the resulting correlation maps over all sub-regions. Temporal scales
are measured by auto-correlations at a range of lags. Further information can be found in Klingaman et al.
(2017) and Martin et al. (2017).

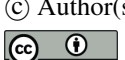



Further, we measure the distribution of the duration of precipitation events in discrete intensity bins by
constructing a two-dimensional (2-D) histogram of binned precipitation intensity against binned duration
in that intensity bin. We calculate the 2-D histogram by aggregating data across the analysis domain, then
normalised by the number of spatial and temporal points in the dataset, to compare across datasets. The
ASoP and duration diagnostics are applied over two subregions of South America: Amazon (AMZ; 10°S –
5°N; 72°W – 50°W) and southeast South America (SESA; 35°S – 18°S; 63°W – 40°W). We apply these
diagnostics to daily data on the native HadGEM3 and CMORPH grids, as well as a common N96 grid.
We produce a 1-D histogram for duration of dry spells, where a dry spell is defined as a time interval of
consecutive precipitation events of less than 0.1 mm.day$^{-1}$. This histogram is normalized by number of
spatial and temporal points in the dataset, to compare across datasets.

## 2.5 Coupling strength metric

Interactions between soil moisture, precipitation, temperature and evaporation modulate climate variability.
We assess the sensitivity of coupling strength between these variables to resolution. Coupling strength is
defined, at each grid point, after removing the linear trend and seasonal cycle, and on the daily time scale,
as
$$r_{a,b}\sigma_b = cor(a,b) \times std(b)$$
Where *cor(a, b)* is the correlation between the variables *a* and *b* and *std* is the standard deviation. As an
example, for the coupling strength between soil moisture (in the top 0.1m of soil) and latent heat flux, *a* is
the soil moisture, and *b* is the latent heat flux. The linear trend was removed over all days, selecting DJF
months only, and across all years to define anomalies relative to the seasonal cycle. We only selected days
over the DJF season, between 1950 and 2014. The coupling strength is also computed with a 2-day lag
correlation.







## 3 Interannual and seasonal means

### 3.1 interannual mean

Observed annual mean precipitation is high over the equator, i.e. the Amazon Basin, Colombia and South Venezuela, while eastern Brazil is relatively dry (Fig. 1a). Precipitation is stronger over the eastern side of the Andes than over the western side, because moisture is carried across South America by the trade easterlies. Over the Andes, peaks in precipitation are collocated with the orography.

HadGEM3 has clear deficiencies in simulating precipitation, particularly over high orography. N96 has a wet bias over southern Brazil and over the Andes, from 30°S to the equator, and a dry bias over north-east Brazil (Fig. 1b). Biases are strong, up to 3 mm.day$^{-1}$ over the Andes. The dry bias over the north-east Brazil is associated with anomalously weak easterlies (Fig. 1b). An anomalously strong cyclonic circulation, located over Peru, weakens the easterlies, between 10°S and the equator, decreasing moisture flux divergence over the western Amazon Basin associated with a wet bias there (Fig. 1b). There is an anomalously strong anticyclonic circulation, over south-east Brazil, which is associated with stronger easterlies from the South Atlantic Ocean to southern Brazil and a wet bias (Fig. 1b).

N216 and N512 also show, wet biases over the Andes and south-eastern Brazil, and dry biases over north-east Brazil (Fig. 1c and Fig. 1d). Biases in low-level winds are also very similar in N96, N216 and N512. We highlight the impacts of each step change in resolution by displaying differences between all pairs of simulations. The total impact of shifting from N96 to N512 is given by N512-N96; intermediate steps are illustrated by N216-N96 and N512-N216. This helps to define the minimum resolution required to extract substantial simulation improvements, from the available sets of simulations. The strongest impact of increasing resolution is over the Andes, where N512-N96 reaches up to 2 mm.day$^{-1}$ (Fig. 2c). Significant differences are also obtained over the Amazon Basin, north-east Brazil and north-west Argentina (Fig. 2a-





c). Over the Amazon basin and the Andes, changes in precipitation in N512-N96 are due to both N216-N96
and N512-N216 (Fig. 2a and Fig. 2b). In addition, differences consist of reduced precipitation (Fig. 2abc),
and thus in reduced wet biases, over the Andes (Fig. 1bcd; see the stippling). Therefore, it is worth
increasing horizontal resolution to N512 for simulating precipitation over the Andes.

Over northern Argentina, significant changes are only due to N216-N96 (Fig. 2a), while there are no
significant changes in N512-N216 (Fig. 2b). Over the Amazon Basin, significant changes are found in both
N216-N96 and N512-N216. Over the Amazon Basin and northern Argentina, increasing resolution
increases precipitation, which strengthens the N96 wet bias. Over north-eastern Brazil, the significant
increase in precipitation with resolution reduces the N96 dry bias. However, the improvement is primarily
found in N216-N96; resolutions higher than N216 do not appear to be useful. Over the Ocean, increased
resolution is associated with strong changes in precipitation, i.e. precipitation increases over the eastern
Pacific Ocean and decreases over the tropical Atlantic Ocean (especially just offshore of most coastal
regions) (Fig. 2), but most of the effect comes from moving from N96 to N216.

Changes in evaporation with resolution are significant over the eastern Pacific Ocean, and over the south-
west Atlantic Ocean, along the coast of South America (Fig. 2d-f). However, increasing resolution leads to
only moderate changes in evaporation over land. Unlike evaporation, differences in moisture flux
convergence (i.e. precipitation minus evaporation) are strong over both land and ocean (Fig. 2g-i).
Therefore, the sensitivity of Amazon Basin and Andes precipitation to resolution is mostly due to sensitivity
in moisture transport rather than in local moisture recycling (i.e. conversion of local evaporation into
precipitation). This is consistent with Vannière et al. (2019), which showed that ocean-to-land moisture
advection increases with resolution. We show small changes in specific humidity and surface air
temperature over land (Fig. S1 and Fig. S2). This suggests that changes in precipitation with resolution are
due to dynamic changes, rather than thermodynamic changes. Increased resolution is associated with an
eastward shift, toward the coast, of the south-east Pacific anticyclonic circulation (Fig. 2g-i) in the southern





Pacific coastal region. The wind speed then strengthens and increases evaporation (Fig. 2d-f) and decreases
moisture convergence (Fig. 2g-i). Over land, changes in wind speed are particularly strong over the
mountains.
**3.2 Seasonal means**
We next examine the influence of resolution on seasonal rainfall, motivated by the strong seasonal cycle of
South American rainfall (i.e., heavy rainfall over northern South America in July-September, while the
Amazon basin is wetter in DJF than in JAS). Over north-east Brazil, the resolution sensitivity is strongest
in DJF and MAM, mainly due N216-N96 (Fig. 3a; Fig. 3c; Fig. 3d and Fig. 3f), while the N512-N216
differences are moderate (Fig. 3b and Fig. 3e). Differences are also strong over the Amazon Basin, in DJF
and SON, where increased resolution increases mean precipitation (Fig. 3c and Fig. 3l). Changes in Amazon
Basin precipitation are contributed by both N216-N96 (Fig. 3a and Fig. 3j) and N512-N216 (Fig. 3b and
Fig. 3k).
Over south-western Brazil—northern Argentina, increasing resolution increases precipitation in all seasons
which increases the wet bias. These changes are only due to N216-N96 (Fig. 3).  Strong differences are also
obtained over the tropical Pacific and Atlantic Ocean, from March to November (Fig. 3d, Fig. 3g and Fig.
3j), mainly due to N216-N96. N512-N216 does not strongly affect oceanic precipitation (Fig. 3e, Fig. 3h
and Fig. 3k).
Improvements are shown over north-east Brazil in DJF and MAM. There is little sensitivity to resolution
elsewhere in South America. Over the Amazon, changes are stronger in austral summer (i.e. DJF), during
the monsoon, but biases are higher at high resolution.




## 4. Seasonal to interannual variability and teleconnections


We have shown a limited effect of resolution on mean precipitation. However, climate variability could be
more sensitive to resolution because resolution may affect how the model simulates precipitation
distribution, local and large-scale atmospheric dynamics, land-atmosphere coupling and mesoscale
systems. Assessing climate variability provides useful information on the ability of climate models to
simulate the climate system.
The pattern in annual precipitation variance follows the pattern in annual mean precipitation, i.e. higher
along the equator than over the surrounding regions (Fig. 4a). At all resolutions, HadGEM3 overestimates
precipitation variability over south-east Brazil, and underestimates precipitation variability between 15°S
and the equator (Fig. 4b-d). HadGEM3 overestimates both mean precipitation and precipitation variability
over parts of the Andes and south-east Brazil/northern Argentina (Fig. 1b-d and Fig. 4b-d). HadGEM3 has
a mean wet bias but underestimates the precipitation variability over the Amazon Basin, although increasing
resolution reduces the variability bias (Fig 4.e-g). Over south-east Brazil, increasing resolution slightly
reduces the overestimation of precipitation variance (Fig. 4e-g). There are no changes in precipitation
variance over north-east Brazil, in N512-N96 (Fig. 4e, Fig. 4f and Fig. 4g).
Precipitation variance also increases with resolution for individual seasons (not shown). Because both
Pacific and Atlantic SSTs affect seasonal-to-interannual South American precipitation variability, we
hypothesized that changes in variance to be associated with a change in the strength of the teleconnection
between ENSO and South American precipitation, and between the South Atlantic SSTs and South
American precipitation. However, this hypothesis was not supported by the following evidences: The
impact of ENSO on South America is assessed through regressing the El Niño 3.4 index (170-120°W; 5°S-
5°N) onto precipitation for each grid point, focusing on the seasonal anomalies (Fig. S3). We found that
increasing horizontal resolution does not systematically alter the influence of ENSO on Brazilian



precipitation. These analyses were repeated, focusing on tropical Atlantic gradients in SST, yielding a
similar conclusion to the one for ENSO, i.e. increasing the horizontal resolution does not change impacts
of the SST on precipitation over land (not shown).



## 5. Daily to sub-seasonal variability and teleconnections

### 5.1 Daily variability

Daily precipitation variance is more sensitive to resolution that monthly or annual variance. Over the Amazon Basin, differences between the simulations are stronger in austral summer than other seasons (Fig. S4). Besides, precipitation variability is strongly tied to the South American summer monsoon, which mainly occurs in DJF. Therefore, we focus further analysis on daily variance and on DJF.

In DJF, N96 underestimates daily precipitation variance (Fig. 5a). N216 and N512 outperform N96, with a reduced underestimation of precipitation variance over the Amazon Basin (Fig. 5b and Fig. 5c). The increase in variance is due to shifts from N96 to N216 and N216 to N512 (Fig. 5d and Fig. 5e). The difference in P-E variance is high, close to the difference in P variance (Fig. 5g; Fig. 5h and Fig. 5i). Therefore, changes in precipitation variance are mostly associated with changes in the variance of moisture flux convergence.

Biases in DJF daily precipitation variance have also been assessed using CMORPH over 1998-2014. The same conclusions are drawn: N96 underestimates variance and N512 overestimates variance (Fig. S4). However, the N96 biases are much reduced when compared to CMORPH instead of GPCC, such that N96 outperforms N216 and N512 (Fig. S4 and Fig. S5). In addition, the northern Brazil circulation is dominated by easterlies (Fig. 1a), whose variability reinforces by increasing the horizontal resolution (Fig. S6). Over southern Brazil, the circulation is dominated by northerlies; increasing resolution increases meridional wind variance (Fig. S7). Therefore, we suggest the change in precipitation variance is associated with changes in atmospheric dynamics. A positive feedback exists since an increase in precipitation is associated with a strengthening of local vertical velocity, which strengthens the low-level wind. However, changes in wind variance exhibit a large-scale pattern that suggests changes that are not due solely to local precipitation increases. The variance of the meridional wind increases strongly over the eastern side of the Andes (Fig. S7), highlighting the importance of the orography in modulating the circulation and transporting moisture.





We analyzed the variance of the zonal and meridional components of the moisture flux and found the same
patterns as for the low-level wind (not shown), suggesting that changes are mostly attributed to dynamic
changes, rather than thermodynamic changes.

## 5.2 Effects of the Madden-Julian Oscillation


The Madden Julian Oscillation (MJO) strongly affects sub-seasonal precipitation variability over Brazil
(Grimm and Silva Dias 1995; Marengo et al. 2008, 2011, 2013; Grimm and Tedeschi 2009; Lewis et al.
2011; Grimm 2019). Therefore, a change in the MJO teleconnection to South America may alter
precipitation mean and variance.
Indices of the Madden-Julian Oscillation (MJO) have been computed using NCEP for observed wind and
outgoing longwave radiation from NOAA Cooperative Institute for Research in Environmental Sciences
data set (Liebmann and Smith 1996), following Wheeler and Hendon (2004), by computing empirical
orthogonal functions on daily values of 850 and 200 hPa zonal winds and outgoing longwave radiation.
Simulated MJO indices are performed by projecting model data onto the reanalysis EOFs, after first
removing the model annual mean and the first three harmonics of the model annual cycle. MJO indices
were computed on data first interpolated on a 2.5° resolution. See Wheeler and Hendon (2004) for a longer
description of the method. Time series have been deseasonalised and linearly detrended prior to computing
impacts of MJO on precipitation mean and variance.
In observations (GPCC), the MJO strongly impacts tropical South American precipitation, leading to above
average precipitation during phases 1 and 8, while phases 3, 4 and 5 are associated with anomalously dry
conditions (Fig. 6, top two rows), as shown in Grimm (2019). South of 20°S, phases 1, 2 7 and 8 are
associated with anomalously dry conditions and phases 3, 4 and 5 with anomalously wet conditions (Fig.
6, top panel). We select two areas, the Amazon Basin, where differences in precipitation variance between
simulations are strong and East Brazil, which is strongly impacted by the MJO. Note the boxes on Fig. 6a.



Both areas experience above average precipitation during MJO phases 1, 7 and 8, and below average
precipitation during phases 3, 4 and 5 (Fig. 6a-b). HadGEM3 reproduces the impact of MJO on East Brazil
and Amazon Basin precipitation in sign and magnitude (Fig. 6i-j). There are no clear differences between
N96, N216 and N512 simulations, and an impact of the horizontal resolution does not emerge.
We show strong impacts of resolution on precipitation variance in Sect. 5.1. Therefore, we address here
how precipitation variance could be affected by resolution within each MJO phase. Results are given
relative to the variance of the precipitation computed from the full original daily timeseries (with no
selection of any specific MJO phases). Results for precipitation variance differ slightly from those for the
mean precipitation, with for instance a decrease in the variance during phase 1 when mean precipitation is
higher, and stronger during phase 3 when mean precipitation is lower. This difference could also arise from
local differences that could strongly impact the area-average. HadGEM3 simulates well the impact of the
MJO on the precipitation variance, with above average variance during phases 7 and 8 and below average
variance during phases 4 and 5. Unlike the observation, HadGEM3 simulates an increase in the variance of
the precipitation during phase 1 of the MJO. N216 and N512 simulations perform better than N96 for phase
3 of the MJO, since the N96 simulates reduced precipitation variance while the variance is anomalously
high in observation and in the N512 and N216 simulations. However, there is no clear sensitivity of MJO-
related precipitation variance to horizontal resolution.

## 5.3 Land-atmosphere feedback


Soil moisture memory contributes to atmospheric variability and could potentially affect the development
of the South American Monsoon System. Land-atmosphere coupling is particularly strong over South
America (Koster et al. 2004; Sörensson and Menéndez 2011). In this section we assess the sensitivity of
land-atmosphere feedbacks to resolution, using ERA-interim as verifying "observations". The coupling



strength metric is defined as the correlation between two variables, weighted by the standard deviation of
the reference variable (see Sect. 2.4).
Over the Amazon Basin, there is a positive relationship between observed precipitation and observed soil
moisture (Fig. 7a), such that an increase in precipitation is associated with anomalously high soil moisture,
with soil moisture are coincident with changes in precipitation (Fig. 7e). Over the Amazon Basin and in all
HadGEM3 resolutions, the bias in the precipitation—soil moisture coupling strength is small (Fig. 7b-d)
and increase in the resolution does not change precipitation—soil moisture coupling strength (Fig. 7i-k;
Fig. 7l-n), probably because, over the Amazon, the soil is saturated, such that increases in precipitation
variability do not impact soil moisture variability. Soil moisture and evaporation are negatively correlated
in observations, such that increased evaporation decreases soil moisture, over the Amazon Basin (Fig. 8a).
Over the Amazon Basin, there is not a strong lead-lag relationship between soil moisture and evaporation
in observations (Fig. 8e) or in HadGEM3 (Fig, 8f-h). The coupling strength is overestimated in N96 (Fig.
8b) but an increase in resolution reduces this overestimation (Fig. 8c-d and Fig.8f-g). Over the Amazon
Basin, the moisture budget is energy-limited, rather than moisture limited (Cook et al. 2014). Therefore,
we also assessed the coupling strength between temperature and evaporation. An increase in temperature is
associated with increased evaporation (Fig. S8) and thus decreased soil moisture, but, in HadGEM3, this
coupling strength is not sensitive to resolution (Fig. S8). These results are consistent with our previous
results, showing that local recycling plays a moderate role in explaining changes in precipitation variance,
which is mainly associated with change in the moisture convergence variability (Fig. 6), rather than with a
stronger land-atmosphere coupling (Fig. 8).
Outside of the Amazon Basin, the soil moisture-precipitation relationship is positive in both observations
(Fig. 7a) and HadGEM3 (Fig. 7b-d), with precipitation variability leading soil moisture variability (Fig. 7b
and Fig. 7f-h). The increase in soil moisture increases evaporation over eastern Brazil (Fig. 8a). The soil
moisture—evaporation coupling strength is too high in all simulations over north-eastern and eastern Brazil
(Fig. 8b-d), with soil moisture driving evaporation, because evaporation is moisture-limited over north-east





Brazil, with changes in evaporation leading changes in temperature (Fig. S8). The strengths of both
precipitation—soil moisture and soil moisture—evaporation couplings are overestimated in N96 (Fig. 7b
and Fig.8b) over eastern Brazil. Increasing resolution reduces this overestimation (Fig. 7cd; Fig. 7i-k; Fig.
8cd; Fig, 8i-k).

**5.4 Scales of precipitation**

We use the ASoP diagnostics (see section 2.4) to assess daily precipitation features over South America in
HadGEM3, and verify them against CMORPH. We compute the fractional contribution to total CMORPH
precipitation from four precipitation intensity bins, over South America, with a focus over two sub-regions,
the Amazon Basin (AMZ) and southeast South America (SESA). We compare spatial and temporal scales
of precipitation features across datasets for the two subregions. Results are given, separately, for light,
moderate and heavy rainfall events. We focus on the occurrence and duration of dry spells.


**5.4.1 Light precipitation and dry spells**
In CMORPH, light precipitation events (<10 mm.day$^{-1}$) contribute the most of all intensity categories to
total precipitation over most of the Andes and northern and southern South America, the Pacific Ocean and
western Atlantic Ocean (Fig. 9a). N96 underestimates contributions from light precipitation events over the
Andes and south-east Brazil, but overestimates contributions from light precipitation over the Amazon
Basin and eastern Brazil (Fig. 9e). The results are consistent with Seth et al. (2004), which also show an
overestimation of the percentage of light rain events over South America. This bias is reduced by increasing
resolution to N216 and N512 (Fig, 9i-p; Fig. S9).
Figure 10 shows frequencies of precipitation events, as classified by intensity and duration. Results are
shown for two regions: AMZ, where variance is too weak; and SESA, where variance is too high. Over





AMZ and SESA, near-zero precipitation (rainy events of 0.1 – 1 mm.day$^{-1}$) can last for more than 15 days, while events of 1 – 10 mm.day$^{-1}$ can last for up to 4 or 5 days (Fig. 10a and Fig. 10f). Over AMZ, N96 overestimates the frequency of events of 2 to 12 mm.day$^{-1}$ and underestimates the frequency of those of less than 1 mm.day$^{-1}$, compared to CMORPH (Fig. 10b). For SESA, N96 underestimates the frequency of precipitation events of less than 1 mm.day$^{-1}$ and lasting between 1 and 8 days; the model overestimates the frequency of near-zero rainy days, lasting more than 8 days (Fig. 10g). Intensity-duration biases improve with resolution over AMZ (Fig. 10c-10d) and SESA (Fig. 10h-10i). However, the biases worsen with resolution for near-zero precipitation lasting for any duration over AMZ, and for intensities between 1-9 mm.day$^{-1}$ with a duration of 1-5 days over SESA.

In addition to events of less than 10 mm.day$^{-1}$, we assess simulated frequency and duration of dry spells, defined by events of less than 0.1 mm.day$^{-1}$. We create 2-D histograms for duration versus frequency of dry days over AMZ and SESA (Fig. 11). CMORPH shows more frequent short-duration dry spells as compared to HadGEM3 over AMZ at both native (Fig. 11a) and N96 (Fig. 11c) resolutions. Over SESA, CMORPH also generally shows more frequent dry spells for durations longer than 1 day (Fig. 11b, 11d). The sensitivity of dry-spell frequency to model resolution is generally smaller than the model bias. Once all datasets are interpolated to the common N96 resolution, N96 produces longer and more frequent dry spells than N216 and N512, and is closer to CMORPH.

## 5.4.2 Moderate precipitation

Over most other parts of South America (i.e. Amazon and central and eastern Brazil), most of the total precipitation is contributed by light to moderate events (10-40 mm.day$^{-1}$; Fig. 9a-c). Compared to CMORPH, N96 overestimates the contribution from moderate events, to total precipitation, over the Andes and underestimates this contribution over South America outside of the Andes (Fig. 9f, 9g). Although the spatial pattern of biases is similar to N96, biases in contribution from moderate rainfall to total precipitation reduce when increasing resolution (Fig. 9f-j-n and Fig. 9g-k-o; Fig. S9).





Over AMZ and SESA, most precipitation comes from moderate events in both CMORPH and HadGEM3
(Fig. 10b-e). Over AMZ, CMORPH distribution peaks at ~30 mm.day$^{-1}$ (Fig. 10b, 10d), when using the
CMORPH native grid (Fig. 10b), and at ~20 mm.day$^{-1}$ when using the N96 grid (Fig. 10d). At their native
resolutions, N96, N216 and N512 have a primary peak at ~9 mm.day$^{-1}$ and a secondary peak at ~30 mm.day$^{-}$
$^{1}$ (Fig. 10b). On the N96 grid, the secondary peak is removed in N216 and N512. As the fractional
contribution in HadGEM3 peaks at lower intensities for all three resolutions, HadGEM3 overestimates the
contribution from intensities below ~15 mm.day$^{-1}$ and underestimates contribution from intensities above
15 mm.day$^{-1}$ (Fig. 10b). When compared on their native grids, the model biases reduce with resolution over
AMZ. However, once interpolated to N96, N512 has the largest bias in fractional contribution, around the
peak intensity (i.e. at ~10 mm.day$^{-1}$). Over AMZ, N96 underestimates the frequency of events of 12-40
mm.day$^{-1}$ (Fig. 10d and Fig. 12b). Increasing resolution reduces the biases for the frequency of events of
12-25 mm.day$^{-1}$ but leads to an underestimation of precipitation of 30 to 40 mm.day$^{-1}$ (Fig. 10b and Fig.
12c-e). Over SESA, distribution peaks at ~20-30 mm.day$^{-1}$ (Fig. 10c and Fig. 10e). Over SESA, N96
underestimates (overestimates) the frequency of events of 2-20 mm.day$^{-1}$ (20-40 mm.day$^{-1}$) (Fig. 10e; Fig.
12g). These biases are reduced in at N216 and N512 (Fig. 10e; Fig. 12h-j).

## 5.4.3 Heavy precipitation

Parts of the Peruvian Andes, Uruguay and eastern Argentina receive most of their rainfall from heavy events
(>40 mm.day$^{-1}$; Fig. 9d).  N96 overestimates these contributions (>40 mm.day$^{-1}$) over central Brazil, the
eastern Amazon and south-eastern Brazil (Fig. 9h). Like for the light and moderate events, increasing
resolution reduces these biases (Fig. 9h-p and Fig. S9). This suggests that, at higher resolution, HadGEM3
performs better for the frequency of extreme events, such as those that lead to flooding. However, the
improvements primarily come from the increase from N96 to N216, not from N216 to N512 (Fig. S9). In
addition, N96 overestimates the frequency of events > 40 mm.day$^{-1}$ over AMZ and SESA (Fig. 10b; Fig.
10g). Increasing resolution reduces these biases, again mostly due to increase from the N96 to N216



resolution, not from N216 to N512. For AMZ, N512 has a higher bias than N216 for events of 40-90
mm.day$^{-1}$.

### 5.4.4 Temporal and spatial scales


To compare spatial and temporal scales of precipitation features across datasets, we plot correlations as
functions of time (Fig. 13a-d) and distance (Fig. 13e-h) (see section 2.4). Over AMZ, N96 overestimates
the spatial and temporal scales of precipitation events relative to CMORPH, on their native grids (Fig. 13a
and Fig. 13e). However, once CMORPH is interpolated to the N96 grid, N96 simulation underestimates the
spatial scale (and overestimates the temporal scale) of precipitation (Fig 13b and Fig. 13f), highlighting that
results strongly depend on the analysis grid. For SESA, N96 also underestimates the spatial scale and
overestimates temporal scale of precipitation (Fig. 13d-g-h). When considering native grids only, there are
no clear differences between N96 and CMORPH for the spatial extent of precipitation events (Fig. 13c).
On native grids, N96 simulates events with larger spatial scales than N216 and N512 (Fig. 13a). However,
this is mainly due to the coarse N96 grid. While all datasets are interpolated onto the N96 grid, N96 events
are smaller than those in N216 and N512, which show similar scales and are closer to CMORPH (Fig. 13b).
Over SESA, spatial scales are similar in all simulations, on their native grids (Fig. 13c). However, as for
AMZ, at N96 resolution N512 and N216 are closer to CMORPH than to N96 (Fig. 13d). For both AMZ
and SESA, therefore, the spatial features of daily precipitation events are better simulated at higher
resolution.
At all resolutions, precipitation features persist longer than in CMORPH (Fig. 13e-h). Over AMZ and
SESA, biases are lowest in N96, which simulates events that are less persistent than in N216 and N512
(Fig. 13f, Fig. 13h). This bias increases at higher resolution. Therefore, increasing horizontal resolution
does not improve biases in temporal scales of precipitation.

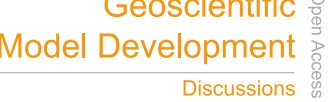
# 6 Conclusion

We assess the effects of increasing horizontal resolution on simulated South American precipitation. We use atmosphere-only simulations, performed with HadGEM3-GC3.1 (Williams et al. 2018) at three horizontal resolutions: N96 (~130 km, 1.88° x 1.25), N216 (~60 km, 0.83° × 0.56°), and N512 (~25 km, 0.35° x 0.23°). We assess, systematically, how the step change between each resolution effects simulated precipitation, focusing on precipitation mean and variance, and on fine scale processes, such as temporal and spatial scales, frequency of heavy and light precipitation events and dry-spell durations.

We show that the atmosphere-only simulations have systematic biases in simulating annual mean and seasonal mean precipitation over South America. North-east Brazil is anomalously dry, while the southeast Brazil and the Andes are too wet. These biases are mostly due to atmospheric circulation biases: underestimated trade easterlies, and a displaced anticyclonic circulation over southeast Brazil, both acting to modify moisture transport over South America. Increasing horizontal resolution affects the simulated precipitation. For instance, precipitation biases reduce over the Andes and over northeast Brazil. It is worth increasing the resolution to N512 (~25 km) for simulating precipitation over the Andes Mountains. This is consistent with Vannière et al. (2019), which shows that the added value of increasing horizontal resolution is greatest over orography. Over northeast Brazil, the largest improvement comes from increasing resolution to N216 (~60 km); a further increase to N512 is only associated with moderate changes. Increasing resolution does not improve model biases over the Amazon Basin. These results are consistent with Roberts et al. (2018) for the Amazon Basin and northeast and south Brazil. In addition, improvements vary seasonally: changes are the strongest over northeast Brazil in DJF and MAM, when precipitation is also highest. Over the Andes, the results are similar in all seasons.

Biases in mean precipitation are collocated with biases in regional precipitation variance. For instance, northeast Brazil is too dry and HadGEM3-GC3.1 systematically underestimates precipitation variance,



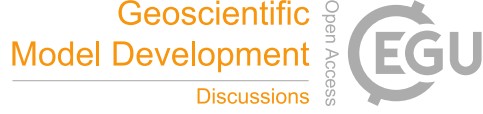

while southeast Brazil is too wet and HadGEM3-GC3.1 systematically overestimates precipitation
variance. However, this does not hold for the Amazon Basin, which is too wet but where the precipitation
variance is strongly underestimated. Precipitation variance is stronger at daily scales than at monthly scales;
biases are strongest in DJF and over the Amazon Basin. Increasing resolution increases precipitation
variance, hence reducing biases. The increase in precipitation variance is associated with an increase in
moisture flux convergence variance over land, and with changes in the variance of the low-level winds;
local recycling of evaporation has a limited role. Relatedly, coupling strengths between evaporation, soil
moisture and precipitation are only weakly sensitive to resolution, except for some improvements in
coupling strength over eastern and south-eastern Brazil. We found only modest sensitivity to resolution for
the teleconnections of the El-Niño Southern Oscillation and Madden-Julian Oscillation to land
precipitation. This suggests that changes in precipitation mean and variance are not due to changes in these
teleconnections.

HadGEM3-GC3.1 has biases in its precipitation distribution. For instance, the model does not produce
enough dry days over the Amazon Basin or moderate rain days (10-40 mm.day$^{-1}$), while simulating too
many light events (<10 mm.day$^{-1}$) and heavy events (>40 mm.day$^{-1}$). Over south-east Brazil, the model
simulates too few short dry spells and too many long ones. HadGEM3-GC3.1 simulates too few and too
short events of 2 to 16 mm.day$^{-1}$, but simulates too many and too long events of more than 20 mm.day$^{-1}$.
These metrics are important for understanding the ability of climate models to simulate high-impact events.
Increasing resolution reduces these biases; N512 is therefore better at simulating precipitation distributions
than N96. In addition, increasing the horizontal resolution increases the spatial scale of daily rain events,
suggesting a better simulation of organised mesoscale systems. However, the persistence of precipitation
events is better simulated at N96, showing no clear sensitivity to resolution. Other models also overestimate
light events at the expense of heavy events over the Amazon and eastern Brazil, and overestimate heavy
events at the expense of lighter ones in southeast Brazil (Seth et al. 2004).





Over South America, precipitation results from the combination of the predominant role played by the
InterTropical Convergence Zone and the South Atlantic Convergence Zone (Waliser et al. 1993; Liebmann
et al. 1999). In addition, mesoscale systems such as squall lines may be responsible for a large fraction of
Amazonian precipitation (Cohen et al. 1995). Our results show that increasing the horizontal resolution
increases the spatial scale of rain events, i.e. of the mesoscale systems, over both Amazonia and south-east
Brazil. Therefore, we speculate that increasing resolution could lead to more organized convective systems,
which would be consistent with the increase in moisture flux convergence, as shown over South America
at the highest resolution. This would be consistent with Vellinga et al. (2016) who showed that N512
resolution improved mesoscale systems over West Africa relative to N96 or N216. Conversely, the decrease
in the persistence of such events (highest at the N96 resolution) could be associated with an increase in
daily rainfall variability, because of less persistent rainy events. Those are hypotheses that should be
assessed in more detail in a specific study, potentially with models at sufficiently high resolution to disable
convective parameterisations.

The mechanism for increases in precipitation variance with resolution are still unclear. The increase in
precipitation variance is a global feature, not limited to South America (Fig. S10). Further work is needed
to understand better this behavior at global scale. Besides, we used AMIP-type simulations; and results
could be different in coupled models, in which the ocean can interact with atmospheric variability,
particularly when accounting for SST teleconnections.











**Code availability**. Codes used to perform analysis and figures are publicly available at

https://doi.org/10.5281/zenodo.3840095. For the analysis of the scales of precipitation (ASoP), codes are

available on https://github.com/nick-klingaman/dubstep/tree/master/asop and https://github.com/nick-

klingaman/dubstep/tree/master/asop_duration.

**Data availability**. The model data used in the analysis are available from the CMIP6 Earth System Grid

Federation, for N96 (HadGEM3-GC31-LM), N216 (HadGEM3-GC31-MM) and N512 (HadGEM3-

GC31-HM). The list of persistent identifiers of the data we have used is available at

**https://doi.org/10.5281/zenodo.3840095**

**Author contributions**. AC, PAM and PC performed the data analysis. PAM prepared the manuscript

with contributions from all co-authors.

**Acknowledgements**

This work was supported by the Newton Fund through the Met Office Climate Science for Service

Partnership Brazil (CSSP Brazil). NPK was funded by an Independent Research Fellowship from the

Natural Environment Research Council (NE/L010976/1) and by the NERC/GCRF programme

Atmospheric hazard in developing countries: risk assessment and early warnings (ACREW). Detailed

calculations and code for the ASoP diagnostics are available at

https://github.com/achevuturi/asop_duration. NOAA OLR data can be obtained from the website

(https://www.esrl.noaa.gov/psd/data/gridded/data.interp_OLR.html).





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



**Figures**

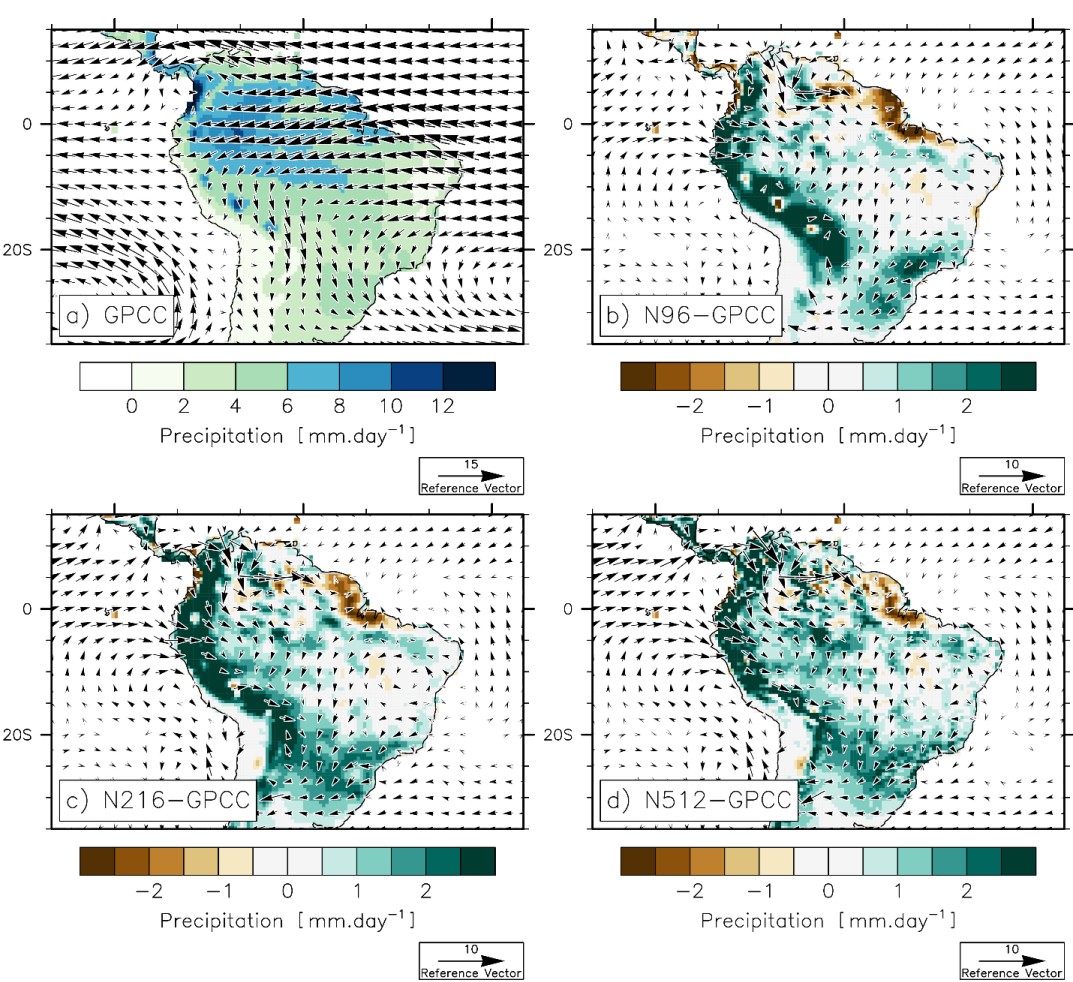


Figure 1: (a) Observed mean annual precipitation (GPCC; mm.day⁻¹; colors) and 850 hPa wind (NCEP;
m.s⁻¹; vectors), averaged over the period 1950-2014. Bias in precipitation and 850 hPa wind in (b) N96 (i.e.
N96-GPCC), (c) N216 (i.e. N216-GPCC) and (d) N512 (i.e. N512-GPCC). On the panels (a), (b) and (c)
biases in precipitation are shown when statistically significant in all of the three members, according to a
Student's t-test and a 95% confidence level.







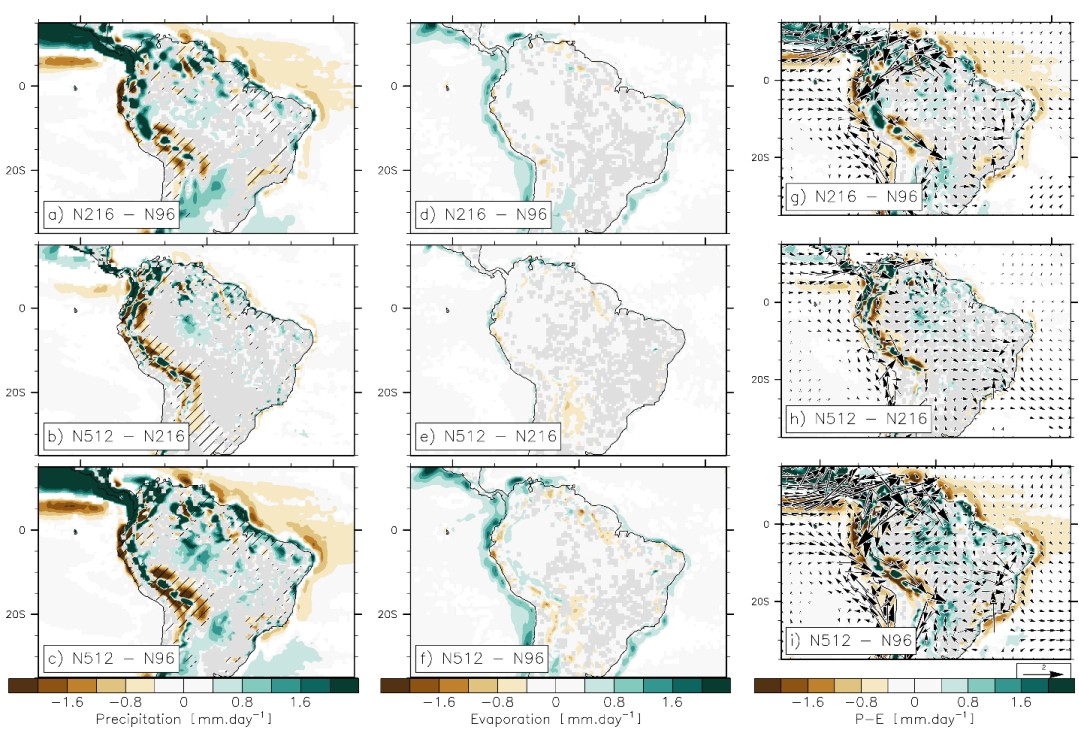


Figure 2: Ensemble-mean (a) N216-N96, (b) N512-N216 and (c) N512-N96 differences in mean annual precipitation (mm.day$^{-1}$). (d), (e) and (f): same as (a), (b) and (c) but for evaporation (mm.day$^{-1}$). (g), (h) and (i): same as (a), (b) and (c) but for the moisture flux convergence (P-E; mm.day$^{-1}$; colors) and the 850 hPa wind (m.s$^{-1}$; vectors). For precipitation (i.e. left row) stippling indicates that the mean bias is reduced at the higher than at the lower horizontal resolution. Differences are shown when significantly different to zero according to a Student's t-test and a 95% confidence level.

789







790

Figure 3: Ensemble-mean N216-N96 difference in (a) DJF, (d), MAM, (g) JJA and (j) SON precipitation (mm.day$^{-1}$). (b), (e), (h) and (k), as in (a), (d), (g) and (j) but for N512-N216. (c), (f), (i) and (l), as in (a), (d), (g) and (j) but for N512-N96. Differences are shown when statistically different to zero, according to a Student's t-test and a 95% confidence level.





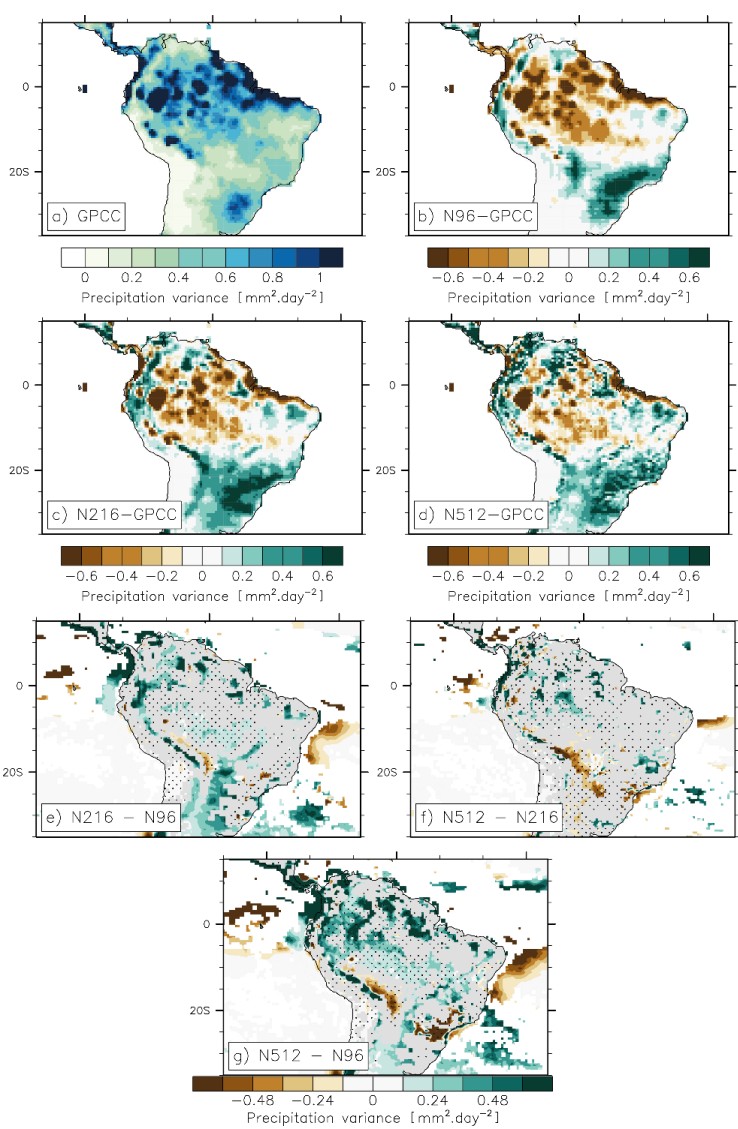


Figure 4: (a) Observed annual-mean precipitation variance (GPCC; mm².day⁻²), as computed over the period 1982-2014. A linear trend is removed. Bias in annual-mean precipitation variance in (b) N96 (i.e. N96-GPCC), (c) N216 (i.e. N216-GPCC) and (d) N512 (i.e. N512-GPCC). (e) N216-N96, (f) N512-N216 and (g) N512-N96 differences in annual-mean precipitation variance. On (b), (c) and (d), biases are shown when all three members produces a bias that is significant according to a f-test and a 95% confidence level. On (e), (f) and (g), stippling indicates that the bias is improved at the higher than at the lower resolution.

806

807

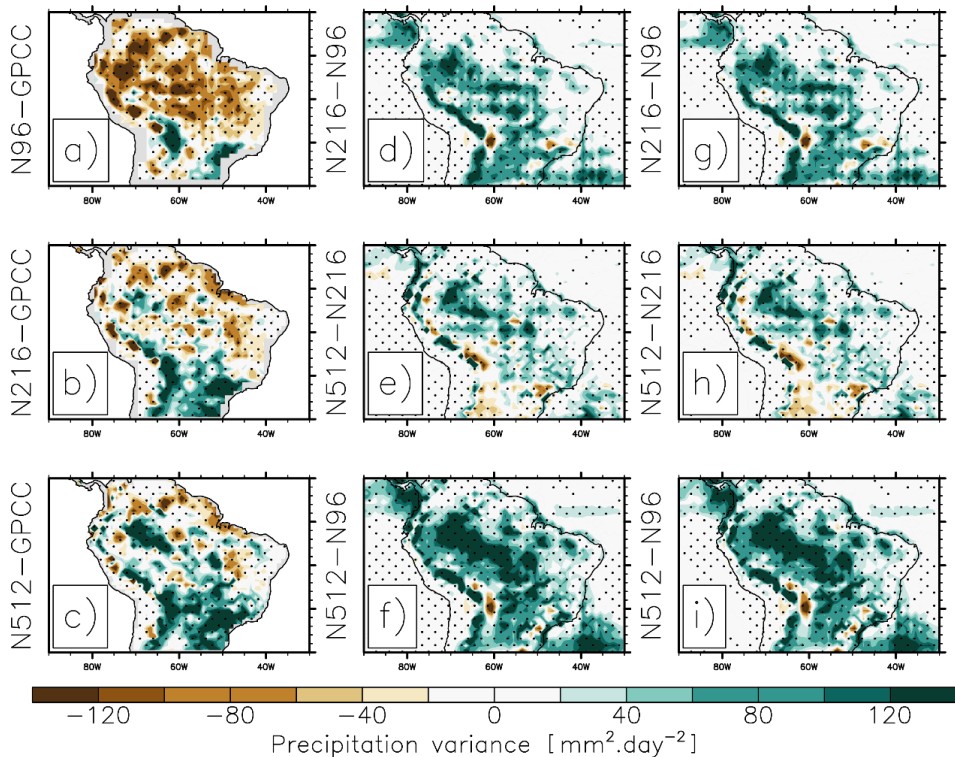

Figure 5: (Left row) Bias in daily precipitation variance (mm$^2$.day$^{-2}$) for (a) N96 (i.e. N96-GPCC), (b) N216 (i.e. N216-GPCC) and (b) N512 (i.e. N512-GPCC) simulations, over the DJF period. Seasonal cycle and linear trend are removed prior to computing variance. Differences in daily precipitation variance (mm$^2$.day$^{-2}$) for (d) N216-N96, (e) N512-N216 and (f) N512-N96. (g), (h) and (i), as in (d), (e) and (f) but for P-E (precipitation minus evaporation) variance.



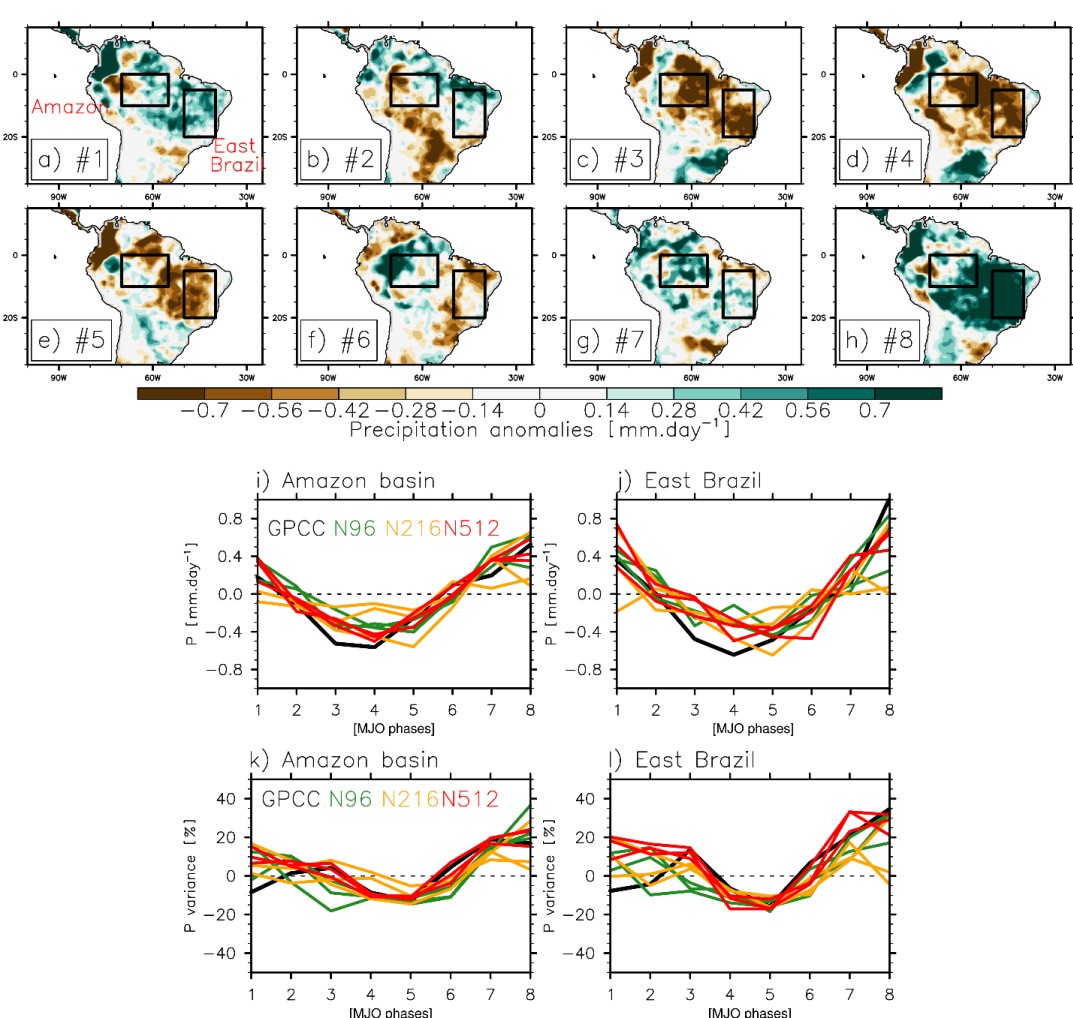

814

Figure 6: Observed impacts of Madden-Julian Oscillation phase (a) 1, (b) 2, (c) 3, (d) 4, (e) 5, (f) 6, (g) 7
and (h) 8 on precipitation (GPCC and NCEP for the RMM index; mm.day$^{-1}$). Precipitation anomalies
(mm.day$^{-1}$), associated with each phase of the Madden-Julian Oscillation, relative to the period 1982-2014,
and averaged over the (i) Amazon Basin and (j) East Brazil (see the box on (a)), for observation (black),
N96 (green), N216 (orange) and N512 (red). (k) and (l), as in (i) and (j) but for precipitation variance, in
percent (%) of the precipitation variance over the period 1982-2014.







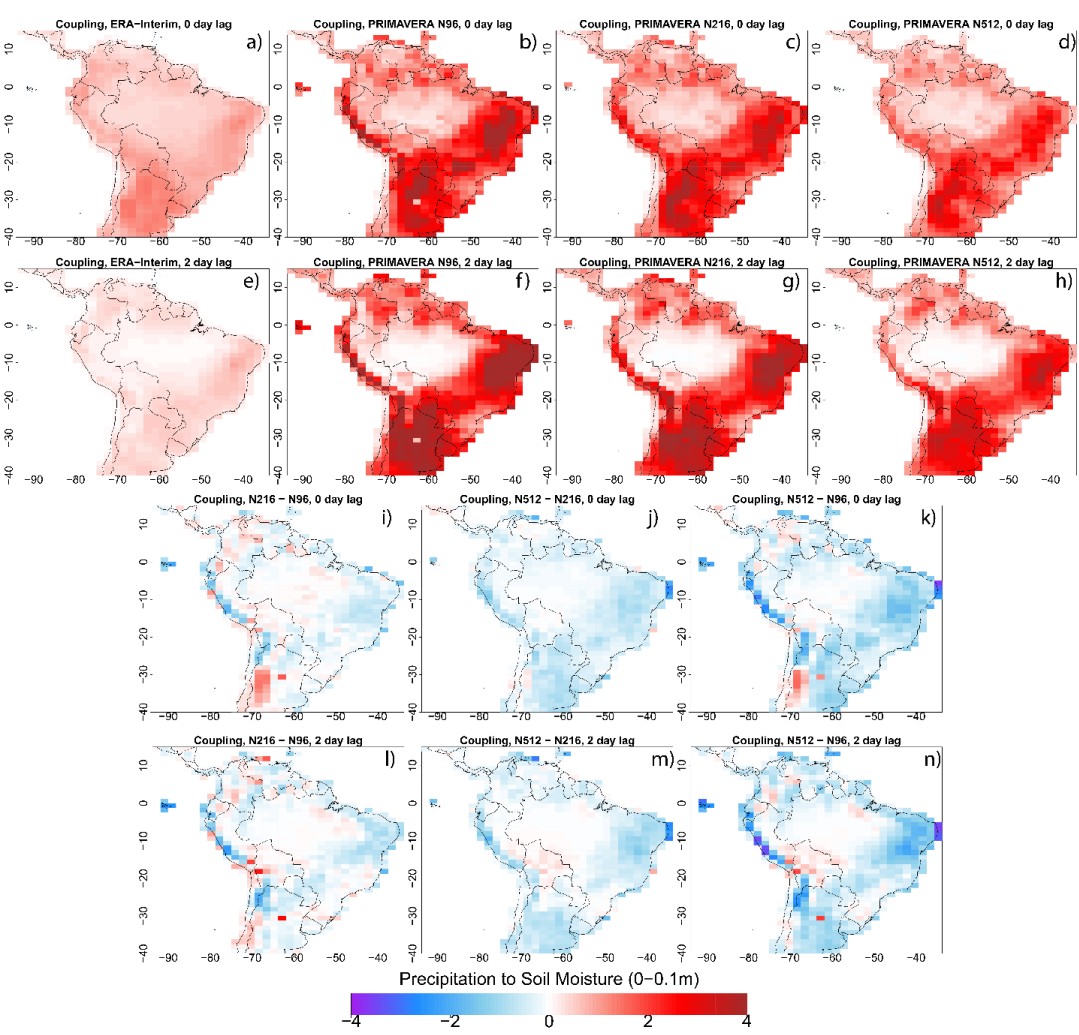


Figure 7: (a) Observed (ERA-Interim) and (b) N96, (c) N216 and (d) N512 Coupling strength ($r_{a,b}\sigma_b$) between daily precipitation and soil moisture (in the top 0.1m of soil) during the southern summer wet season (DJF), over the period 1979-2014. 2-day time lag (i.e. the soil situation 2 days after precipitation) for (e) ERA-Interim, (f) N96, (g) N216 and (h) N512. (i) N216-N96, (j) N512-N216 and (k) N512-N96 coupling strength. (l), (m), (n), as for (i), (j) and (k) but with a 2-day time lag between precipitation and soil moisture.






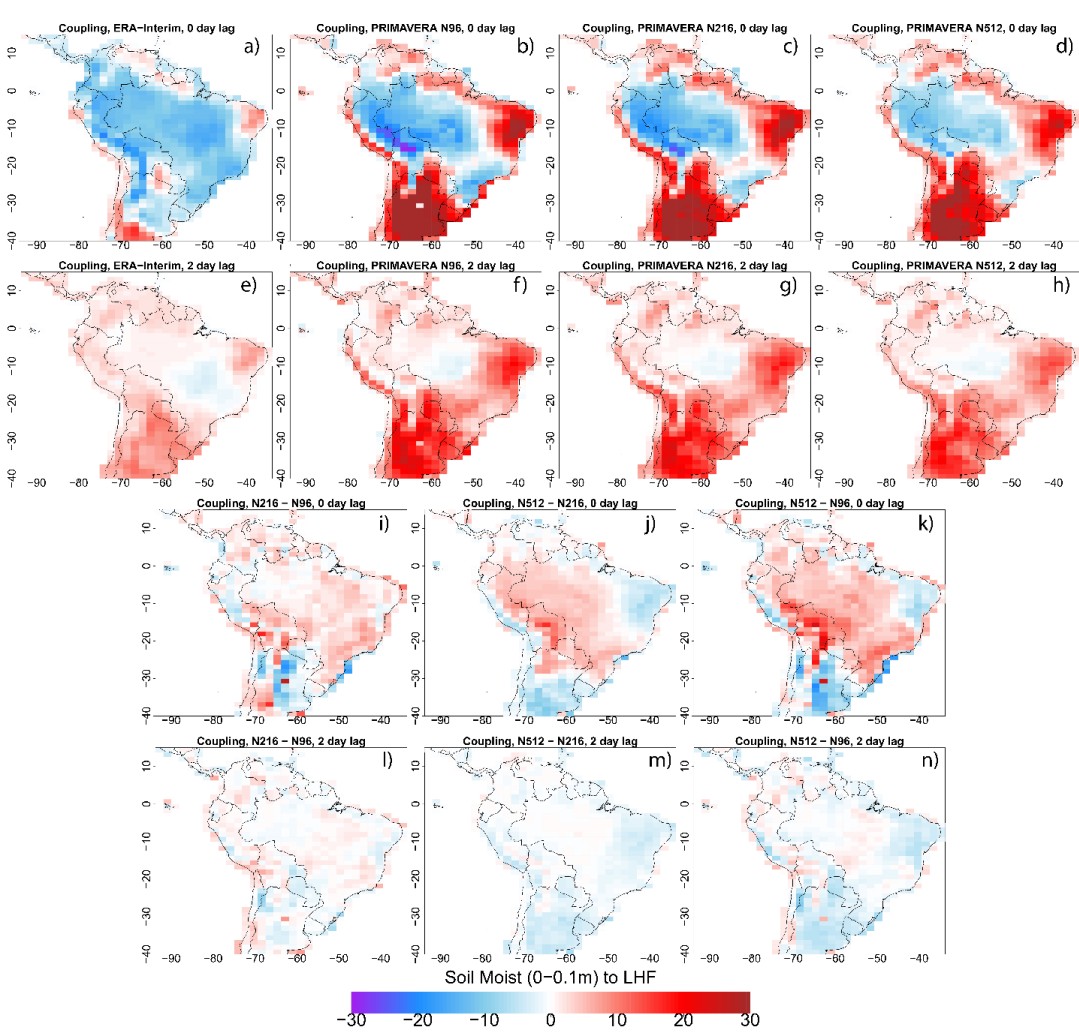


Figure 8: As in Figure 7 but for the coupling strength between daily soil moisture (in the top 0.1m of
soil) and latent heat flux (LHF).

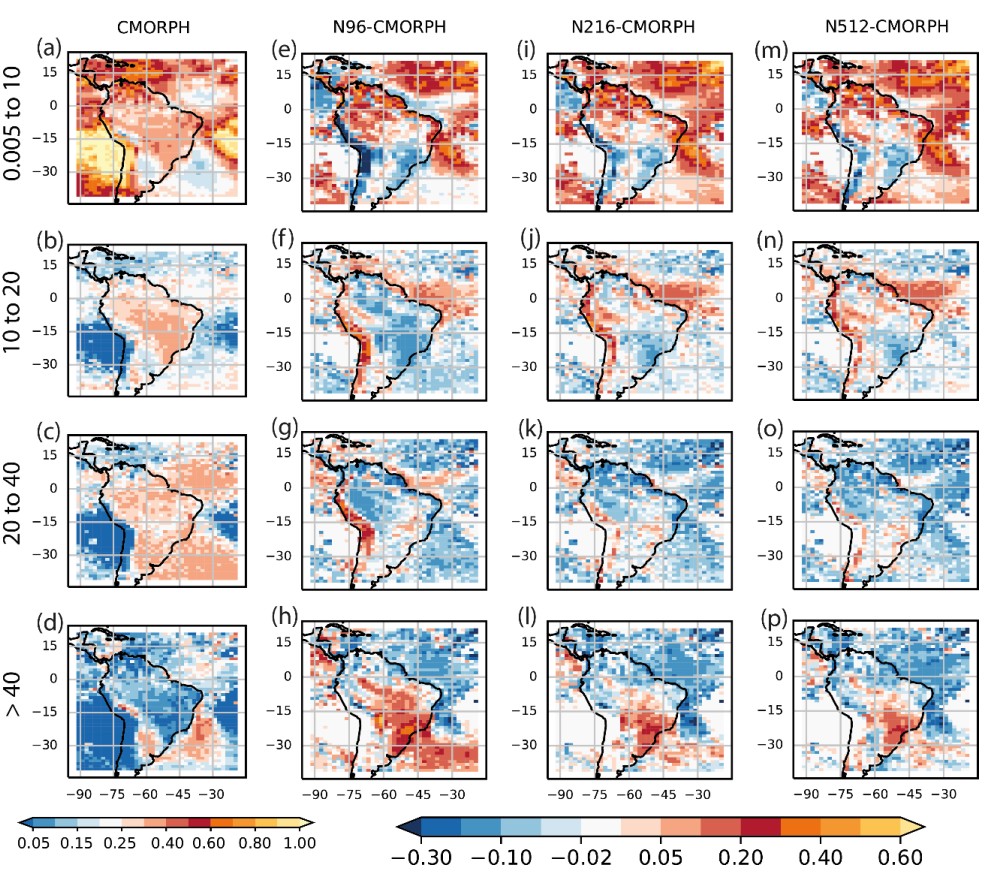

Figure 9: Fractional contribution to the total precipitation from ranges of intensity bins shown in the labels above each panel for CMORPH (a-d) (the sum of each column is unity). Differences in the fractional contributions compared against CMORPH for N96 (e-f), N216 (i-l) and N512 (m-p) all on the N96 common grid. The four ranges of intensity bins are (first row) 0.005 to 10 mm/day, (second row) 10 to 20 mm/day, (third row) 20 to 40 mm/day and (last row) >40 mm.day$^{-1}$.





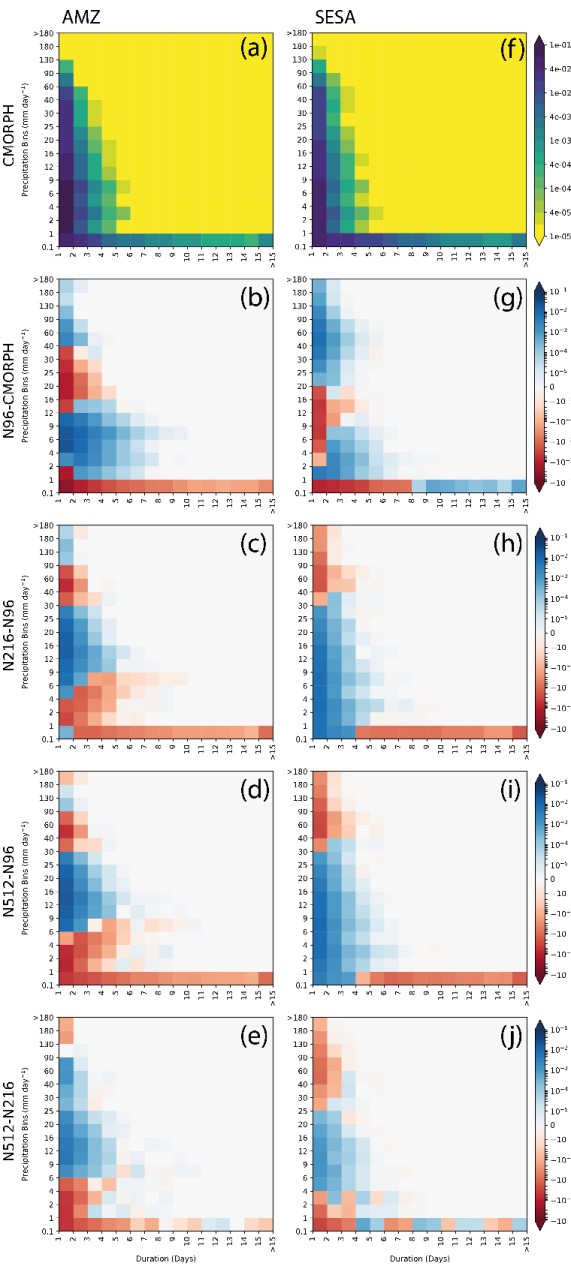

847

Figure 10: Two-dimensional histograms of binned precipitation lasting for each duration bin, aggregated over all grid points and normalized by the number of spatial and temporal points in each dataset for (a) CMORPH for the AMZ region at N96 grid. Differences between the two-dimensional histograms for (b) N96 minus CMORPH; (c) N216 minus N96; (d) N512 minus N96 and (e) N512 minus N216 computed on the common N96 grid. (f-j) is same as (a-e) but for the SESA region.

853

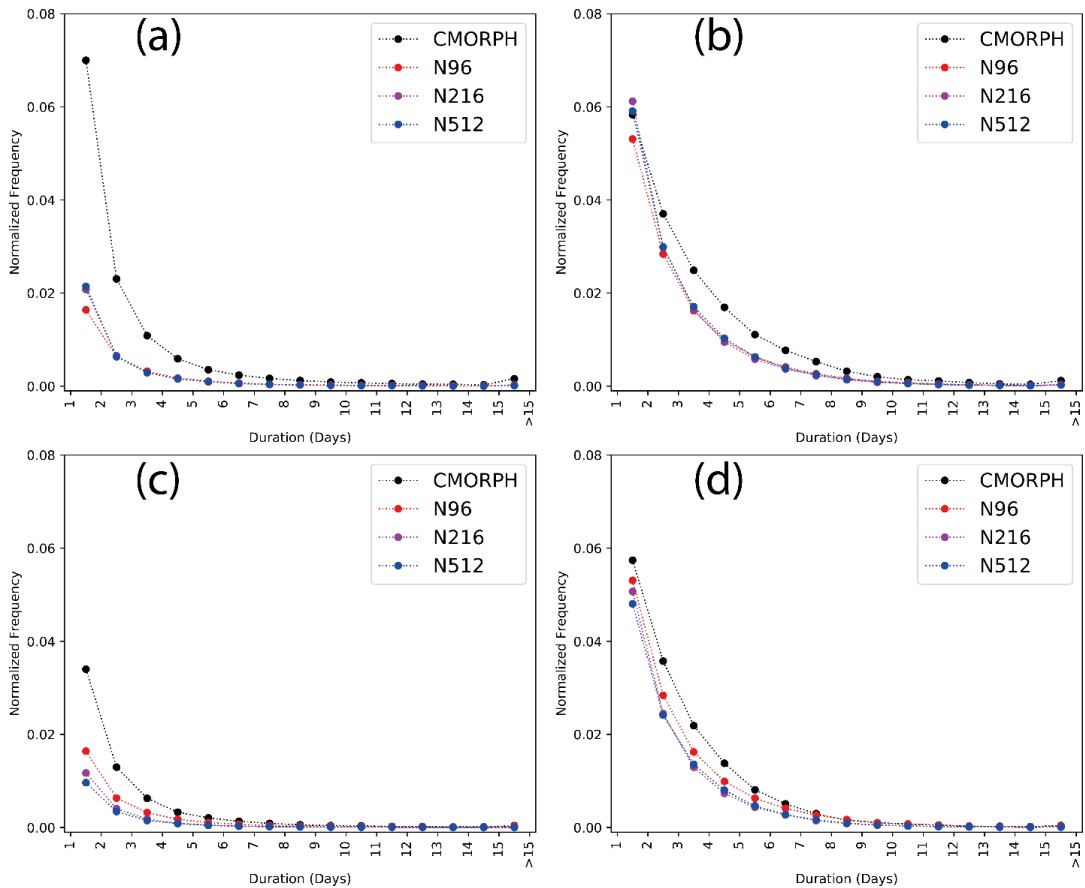

854

Figure 11: Histograms of dry days (with precipitation less than 0.1 mm day$^{-1}$) lasting for each duration bin, aggregated over all grid points and normalized by the number of spatial and temporal points in each dataset (a) Amazon and (b) SESA at native resolution for all datasets. (c-d) is same as (a-b) but for datasets on the common N96 grid.

859

860

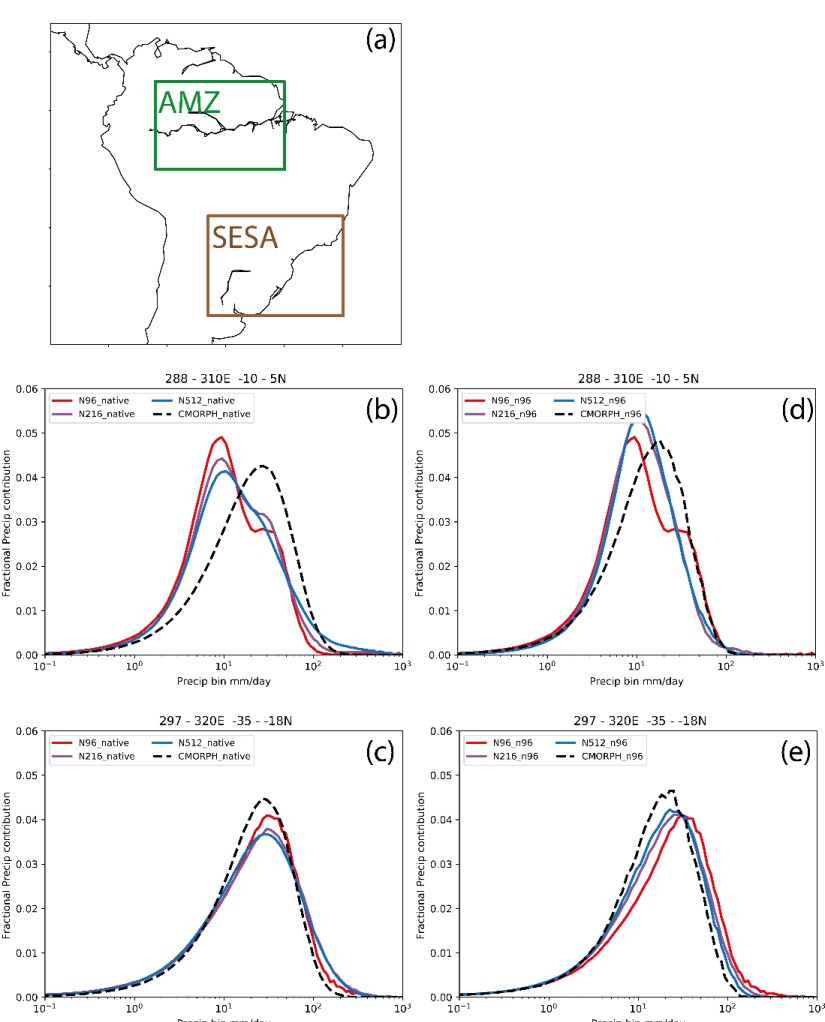

861

Figure 12: (a) Subregions used in our study (i) the Amazon region (AMZ; green box; 10°S – 5°N; 72°W –
50°W) and (ii) the southeast South America region (SESA; brown box; 35°S – 18°S; 63°W – 40°W).
Histograms of the average precipitation contributions to the total precipitation from each precipitation bin
for CMORPH and all simulations on their native grids (b) AMZ and (c) SESA. (d-e) is same as (b-c) but at
96 grid.

867





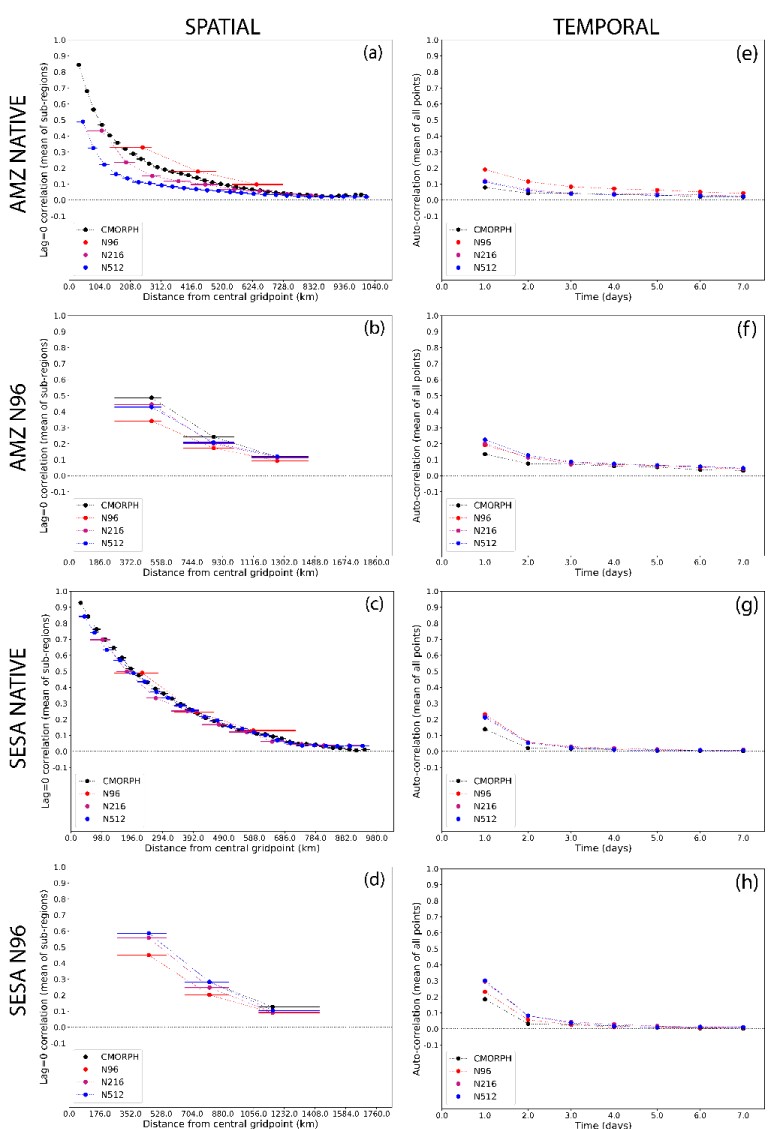

868

Figure 13: (a) metric of the spatial scale of daily precipitation (at native resolution), computed by dividing the analysis domain into 1500 km x 1500 km sub-regions and calculating the mean lag-0 correlation between the central grid point and all grid points within each distance bin (which are 1 delta x wide, starting from 0.51x) away from the central grid point, then averaging the correlations over all sub-regions in AMZ; (e) metric of the temporal scale of daily precipitation, computed as the autocorrelation at each point, averaged over all points AMZ. The horizontal lines in (a-d) show the range of distances spanned by each distance bin; the filled circle is placed at the median distance. For clarity, we omit the correlations for zero distance and zero lag, which are 1.0 by definition. (b and f) same as (a and c) respectively for all datasets on the N96 grid; (c-d and g-h) same as (a-b and e-f) respectively but for SESA.