# Peer review of "Role of atmospheric horizontal resolution in simulating 1 tropical and subtropical South American precipitation in 2 HadGEM3-GC31 3 4 Paul-Arthur Monerie1, Amulya Chevuturi1, Peter Cook1, Nick Klingaman1, Christopher E. Holloway2 5 6 1 Department of Meteorology, National Centre for Atmospheric Science (NCAS), University of 7 Reading, Reading, UK 8 2Department of Meteorology, University of Reading,"

_Geoscientific Model Development, 2020_

## Referee Comment (RC1) · Anonymous Referee #1 · 21 Jul 2020

Review: HadGEM3-GC3.1 atmospheric-only simulations are assessed to discuss the impacts of horizontal resolution increasing on the precipitation climatology and precipitation variabilities (in intensity and in the space and time) over South America. Three ensembles of HadGEM3-GC3.1 with horizontal grid spacing of approximately ~130 (N96), 60 (N216) and 25 km (N512) are compared with reanalysis (NCEP and ERA-Interim) and satellite data (CMORPH) to evaluate the impacts of resolution on precipitation using different metrics (climatology, seasonality, large scale influences of MJO and ENSO, coupling between precipitation and soil moisture, intensity distribution, dry spells, etc.). The results are new and very relevant since are showing that improvements on precipitation occur when the resolution is increased from N96 to N216 for

most regions of South America, while over the Andes Mountains the improvements continue until N512. The improvements are associated with better simulation of moisture flux convergence and daily precipitation distribution at fine resolution. In addition, the authors do not found any relevant impacts of resolution on low-frequency variability of precipitation (MJO and ENSO forcings). Overall, this study contributes to understanding the impacts of model resolution on precipitation at spatial and temporal and some limitation of resolution refinements. The manuscript has new contributions to the atmospheric modeling area and it is worthy of publishing after some minor revisions.

Minor comments

In some parts of the text appear "north-east", "south-east" and in others, respectively, "northeast", "southeast" to refer to the same geographical regions in Brazil ((Lines: 16, 17, 31, 32, 43, . . ., 504, 508, L514, . . .). Please, to unify how to refer to these regions preferentially using "northeast" and "southeast"

L32, L51, L82 – "de Souza Custodio et al. 2017)" to "Custodio et al. 2017)"

L38 – to remove "over South America"

L57 – In relation to the "South American Monsoon System
. . ." to refer to Vera et al. (2006). Vera, C., et al. (2006), A unified view of the American monsoon systems, J. Clim., 19, 4977–5000.

L103 – should be "improves the modeled precipitation variability over . . ."

L142-143 – Please, to include the information of what are the horizontal resolutions of GPCC, University Delaware, NCEP-NCAR and ERA-Interim.

L146 – The citation of ERA-Interim in this context is wrong since it is available only from 1979. Please, check.

L194 – I suggest to change "over the equator . . ." to "over tropical latitudes . . ."\

L195 – "eastern Brazil is relatively dry" should be "northeastern Brazil is relatively dry"

since in subtropical eastern of Brazil precipitation is between 4-6 mm/day, which can not be considered dry.

L231- Is hard to interpret Figures 2d-e-f since they do not show any important difference over the continent. This occurs because they are using the same scale of Figures 2-a-b-c. I suggest to the authors to remove Figures 2d-e-f or to change the scale to illustrates what is important in terms of evapotranspiration over continental areas. L346 – change "1, 7 and 8 . . ." to "1, 7 and 8 (Fig. 6a-g-h) . . ."

L386 – change to "moisture flux convergence . . ."

L395 – "over eastern Brazil . . ." should be "over eastern Brazil and southeastern South America " L416 – I am seeing overestimation in Figure 9e over northeastern Brazil (the box to east 45oW and north 15oS) and not over "eastern Brazil". Please, verify the affirmation.

L457-459 – Please, check the letters of Figures 10 and 12: a) L457 "Fig. 10c and Fig. 10e" should be "Fig. 10h and Fig. 10j"; b) L458 "Fig. 10e; Fig. 12g" should be "Fig. 10g; Fig 12e"; c) L459 "Fig. 10e; Fig. 12h-j" should be "Fig. 10h-j; Fig. 12e".

L461 – The correct location are "Peruvian Andes, Paraguay, and northeastern Argentina"

L475 – "function of time (Fig. 13a-d) and distance (Fig. 13e-h). . ." should be "function of distance (Fig. 13a-d) and time (Fig. 13e-h)"

L489 – "precipitation features . . ." should be "simulated precipitation features . . ."

---

## Referee Comment (RC2) · Anonymous Referee #1 · 9 Aug 2020

The authors have addressed adequately my previous comments. My recommendation is to accept the manuscript for publication.

---

## Referee Comment (RC3) · Anonymous Referee #2 · 10 Aug 2020

This manuscript investigates the impact of increasing horizontal resolution in the climate model HadGEM3-GC31 and how it impacts the model's representation of precipitation in South America. The manuscript evaluates the impact of increasing horizontal resolution on interannual and seasonal precipitation variability; daily precipitation variability; different precipitation intensities; as well as some aspects of remote forcing and local effects. Overall, a very thorough analysis. Moreover, South America (and the southern hemisphere in general) doesn't receive as much attention from the scientific community in comparison to the northern hemisphere. Therefore, this work is relevant to the scientific community with important societal implications for South America.

[Figure]

The manuscript is very well written, the figures have high quality, and I believe the authors already answered most of the concerns from the reviewers. However, I would like to propose just one more discussion, which is related to Line 78 and seasonal precipitation predictability.

Jia et al. (2015) suggests that higher atmospheric and land resolution "can" improve seasonal forecasts when combined with statistical analysis. Bombardi et al. (2019) performed a somewhat similar analysis to this manuscript, but focusing on summer precipitation predictability using the IFS (ECMWF) model. They found no significant improvement in seasonal predictability of summer precipitation due to an increase in resolution. Although there seems to be some value in increasing the resolution of the both the atmosphere and the ocean. There is some consensus in the scientific community that an increase in spatial resolution without an appropriate improvement of model physics does not lead to better forecasts, because the increase in resolution leads to an increase in noise. I don't expect the authors to perform any more analysis, I would just like to ask the authors to perhaps include a discussion on how their findings related to studies that focus on precipitation predictability (e.g. Becker et al. 2014; Jia et al. 2015; Bombardi t al. 2019). The argument here is that an increase in spatial resolution leads to an improvement of the model's representation of precipitation. Right. But ultimately, we want the model to be able to predict precipitation. Considering the computational cost of climate simulations and potentially negative effects of increasing spatial resolution, should we really advocate for simulations to be performed with higher spatial resolution? Just some thoughts on the matter would suffice.

Becker, E., H.van den Dool, and Q.Zhang, 2014: Predictability and forecast skill in NMME. J. Climate, 27, 5891–5906, https://doi.org/10.1175/JCLI-D-13-00597.1.

Bombardi, R. J., L. Trenary, K. Pegion, B. Cash, T. DelSole, and J. L. Kinter, 2018: Seasonal Predictability of Summer Rainfall over South America. J. Climate, 31, 8181–8195, https://doi.org/10.1175/JCLI-D-18-0191.1.

---

## Author Comment (AC2) · 13 Aug 2020

We thank the reviewers for their constructive comments and suggestions. We have given full consideration to the comments in the revised manuscript. We have added a short discussion in the paper, following your suggestion, and have included the citation to Bombardi et al. (2018).

Please see, lines 565-568: "Although we hypothesized that increasing resolution might affect the ability of climate models to predict precipitation, Bombardi et al. (2018) have shown that an improvement of South American precipitation prediction due to an increase in resolution is not straightforward. In addition to resolution, further works

should, therefore, be devoted to understanding the effects of physic, on prediction system performance."

---

## Author Response (AR1)

We thank the reviewers for their constructive comments and suggestions. We have given full consideration to the comments in the revised manuscript. Please find below a point-by-point reply to the questions raised. Please note that in addition to the reviewer comments we have added a sentence in the acknowledgements: "The authors thank the two anonymous reviewers for their constructive comments and suggestions."

**Reviewer #1**:

Review: HadGEM3-GC3.1 atmospheric-only simulations are assessed to discuss the impacts of horizontal resolution increasing on the precipitation climatology and precipitation variabilities (in intensity and in the space and time) over South America. Three ensembles of HadGEM3-GC3.1 with horizontal grid spacing of approximately~130(N96), 60 (N216) and 25 km (N512) are compared with reanalysis (NCEP and ERA-Interim) and satellite data (CMORPH) to evaluate the impacts of resolution on precipitation using different metrics (climatology, seasonality, large scale influences of MJO and ENSO, coupling between precipitation and soil moisture, intensity distribution, dry spells, etc.). The results are new and very relevant since are showing that improvements on precipitation occur when the resolution is increased from N96 to N216 for most regions of South America, while over the Andes Mountains the improvements continue until N512. The improvements are associated with better simulation of moisture flux convergence and daily precipitation distribution at fine resolution. In addition, the authors do not found any relevant impacts of resolution on low-frequency variability of precipitation (MJO and ENSO forcings). Overall, this study contributes to under-standing the impacts of model resolution on precipitation at spatial and temporal and some limitation of resolution refinements. The manuscript has new contributions to the atmospheric modeling area and it is worthy of publishing after some minor revisions.

Minor comments

In some parts of the text appear "north-east", "south-east" and in others, respectively, "northeast", "southeast" to refer to the same geographical regions in Brazil ((Lines: 16,17, 31, 32, 43,..., 504, 508, L514,...). Please, to unify how to refer to these regions preferentially using "northeast" and "southeast"

Thank you for your comment, we have rephrased the text, using northeast and southeast instead of north-east and south-east.

L32, L51, L82 – "de Souza Custodio et al. 2017)" to "Custodio et al. 2017)"

We have changed the reference throughout text, editing the reference to Custodio et al. (2017).

L38 – to remove "over South America"

We have removed "over South America"

L57 – In relation to the "South American Monsoon System ǎ́Á́Í..." to refer to Vera et al.(2006). Vera, C., et al. (2006), A unified view of the American monsoon systems, J.Clim., 19, 4977–5000.

Thank you for the reference, we have now added Vera et al. (2006) in the main text.

L103 – should be "improves the modeled precipitation variability over..."

We have changed the sentence accordingly to your comment.

L142-143 – Please, to include the information of what are the horizontal resolutions of GPCC, University Delaware, NCEP-NCAR and ERA-Interim.

Both GPCC and UDEL precipitation are provided on a 0.5° horizontal resolution. NCEP-NCAR is gen at a 2.5° horizontal resolution and ERA-interim at a 1.5° horizontal resolution. This information has been added to the data section (Sect. 2.2). Please see: "To evaluate time-mean rainfall and sub-seasonal to seasonal variability, we compare HadGEM3 to longer-period, but lower-resolution, gauge-based datasets from the University of Delaware (Willmott et al. 2001) and from the Global Precipitation Climatology Centre (GPCC; Schneider et al. 2014), both at a 0.5° horizontal resolution. We assess mean circulation against the NCEP-NCAR reanalysis (Kanamitsu et al. 2002), given on a 2.5° resolution (144 × 72) with 17 vertical levels, and ERA-interim reanalysis (Dee et al. 2011), given on a 1.5° horizontal resolution."

L146 – The citation of ERA-Interim in this context is wrong since it is available only from 1979. Please, check.

Thank you for your comment. This is a mistake, we have used NCEP to assess biases in monthly mean wind. The sentence has been corrected.

L194 – I suggest to change "over the equator..." to "over tropical latitudes..."\

We have changed "over the equator" to "over tropical latitudes".

L195 – "eastern Brazil is relatively dry" should be "northeastern Brazil is relatively dry" since in subtropical eastern of Brazil precipitation is between 4-6 mm/day, which cannot be considered dry
Thank you for you comment, we have rephrased the text following your suggestion.
L231- Is hard to interpret Figures 2d-e-f since they do not show any important difference over the continent. This occurs because they are using the same scale of Figures2-a-b-c. I suggest to the authors to remove Figures 2d-e-f or to change the scale to illustrates what is important in terms of evapotranspiration over continental areas.

Our point is here to show that effect of the resolution it not mediated by changes in evaporation, and that effect on precipitation is primary due to large-scale changes rather than to local changes. Therefore, we think that it is important to keep the changes in evaporation in the main text. We do prefer to keep the same scale to compare changes in moisture flux convergence and evaporation so that both fields are easily comparable. We agree that patterns in evaporation but this is due to the fact that changes in evaporation are rarely significant and that changes in precipitation are most only due to changes in moisture flux convergence.

L346– change "1, 7 and 8..." to "1, 7 and 8 (Fig. 6a-g-h)..."

We have corrected the typo.

L386 – change to "moisture flux convergence..."

We have rephrased the sentence, using "moisture flux convergence" instead of "moisture convergence".

L395 – "over eastern Brazil..." should be "over eastern Brazil and southeastern South America "

Thank you for your comment, we have added "and southeastern South America" in the sentence.

L416 – I am seeing overestimation in Figure 9e over northeastern Brazil (the box to east 45oW and north 15oS) and not over "eastern Brazil". Please, verify the affirmation.

We have rephrased the sentence, changing "eastern Brazil" by "northeastern Brazil".

L457-459 – Please, check the letters of Figures 10 and 12: a) L457 "Fig. 10c and Fig.10e" should be "Fig. 10h and Fig. 10j"; b) L458 "Fig. 10e; Fig. 12g" should be "Fig.10g; Fig 12e"; c) L459 "Fig. 10e; Fig. 12h-j" should be "Fig. 10h-j; Fig. 12e".

We have rephases the text, correcting "Fig. 10c and Fig. 10e" by "Fig. 12c and Fig. 12e", "Fig. 10e; Fig. 12g" by "Fig.10g; Fig 12e", and "Fig. 10e; Fig. 12h-j" by "Fig. 10h-j; Fig. 12e".

L461 – The correct location are "Peruvian Andes, Paraguay, and northeastern Argentina"
We have rephrased the sentence, changing "eastern Argentina" by "northeastern Argentina".

L475 – "function of time (Fig. 13a-d) and distance (Fig. 13e-h)..." should be "function of distance (Fig. 13a-d) and time (Fig. 13e-h)"

Thank you to pointing this mistake out. We have corrected the text.

L489 – "precipitation features..." should be "simulated precipitation features..."

We have rephrased the text following your suggestion.

**Reviewer #2**:

The manuscript is very well written, the figures have high quality, and I believe the authors already answered most of the concerns from the reviewers. However, I would like to propose just one more discussion, which is related to Line 78 and seasonal precipitation predictability.

Jia et al. (2015) suggests that higher atmospheric and land resolution "can" improve seasonal forecasts when combined with statistical analysis. Bombardi et al. (2019) performed a somewhat similar analysis to this manuscript, but focusing on summer precipitation predictability using the IFS (ECMWF) model. They found no significant improvement in seasonal predictability of summer precipitation due to an increase in resolution. Although there seems to be some value in increasing the resolution of the both the atmosphere and the ocean. There is some consensus in the scientific community that an increase in spatial resolution without an appropriate improvement of model physics does not lead to better forecasts, because the increase in resolution leads to an increase in noise. I don't expect the authors to perform any more analysis, I would just like to ask the authors to perhaps include a discussion on how their findings related to studies that focus on precipitation predictability (e.g. Becker et al. 2014; Jia et al. 2015; Bombardi t al. 2019). The argument here is that an increase in spatial resolution leads to an improvement of the model's representation of precipitation. Right. But ultimately, we want the model to be able to predict precipitation. Considering the computational cost of climate simulations and potentially negative effects of increasing spatial resolution, should we really advocate for simulations to be performed with higher spatial resolution? Just some thoughts on the matter would suffice.

Becker, E., H.van den Dool, and Q.Zhang, 2014: Predictability and forecast skill in NMME. J. Climate, 27, 5891–5906, https://doi.org/10.1175/JCLI-D-13-00597.1.

Bombardi, R. J., L. Trenary, K. Pegion, B. Cash, T. DelSole, and J. L. Kinter, 2018: Seasonal Predictability of Summer Rainfall over South America. J. Climate, 31, 8181– 8195,

We thank the reviewers for their constructive comments and suggestions. We have given full consideration to the comments in the revised manuscript. We have added a short discussion in the paper, following your suggestion, and have included the citation to Bombardi et al. (2018).

[revised manuscript text omitted]

---

## Author Response (AR2)

Response to the topical editor.

We have added a repository in Zenodo, as asked, please see the code availability section, "and https://zenodo.org/record/3997114#.X0PGrjV7mUk." In addition, we have updated the style of the references to be consistent with GMD.